# Detection of porcine enteric viruses (Kobuvirus, Mamastrovirus and Sapelovirus) in domestic pigs in Corsica, France

**Lisandru Capai**[1]*, **Géraldine Piorkowski**[2], **Oscar Maestrini**[3], **François Casabianca**[3], **Shirley Masse**[1], **Xavier de Lamballerie**[2], **Rémi N. Charrel**[2], **Alessandra Falchi**[1]*

**1** UR 7310, Laboratoire de Virologie, Université de Corse, Corte, France, **2** Unité des Virus Émergents (UVE: Aix-Marseille Univ-IRD 190-Inserm 1207), Marseille, France, **3** Laboratoire de Recherche sur le Développement de l'Elevage (LRDE), Institut National de Recherche pour l'Agriculture, l'Alimentation et l'Environnement (INRAE), Corte, France

\* capai_l@univ-corse.fr (LC); falchi_a@univ-corse.fr (AF)

## Abstract

Many enteric viruses are found in pig farms around the world and can cause death of animals or important production losses for breeders. Among the wide spectrum of enteric viral species, porcine Sapelovirus (PSV), porcine Kobuvirus (PKoV) and porcine Astrovirus (PAstV) are frequently found in pig feces. In this study we investigated sixteen pig farms in Corsica, France, to evaluate the circulation of three enteric viruses (PKoV, PAstV-1 and PSV). In addition to the three viruses studied by RT–qPCR (908 pig feces samples), 26 stool samples were tested using the Next Generation Sequencing method (NGS). Our results showed viral RNA detection rates (i) of 62.0% [58.7–65.1] (n = 563/908) for PSV, (ii) of 44.8% [41.5–48.1] (n = 407/908) for PKoV and (iii) of 8.6% [6.8–10.6] (n = 78/908) for PAstV-1. Significant differences were observed for all three viruses according to age (P-value = 2.4e–13 for PAstV-1; 2.4e–12 for PKoV and 0.005 for PSV). The type of breeding was significantly associated with RNA detection only for PAstV-1 (P-value = 9.6e–6). Among the 26 samples tested with NGS method, consensus sequences corresponding to 10 different species of virus were detected. This study provides first insight on the presence of three common porcine enteric viruses in France. We also showed that they are frequently encountered in pigs born and bred in Corsica, which demonstrates endemic local circulation.

## Introduction

Pig farms make an important contribution to the economy of world agriculture and are an important source of food. Porcine diarrhea can cause mortality in animals, especially in piglets, and cause economic losses to the pig farmers; many of the pathogens responsible can also infect humans. A very broad spectrum of viruses that can cause porcine diarrhea has been found in pig feces, including porcine Sapelovirus (PSV), porcine Kobuvirus (PKoV), porcine Sapovirus, porcine Astrovirus (PAstV), porcine Bocavirus and porcine Rotavirus [1–7]. The

**Data Availability Statement:** All relevant data are within the manuscript and its Supporting Information files.

**Funding:** The author(s) received no specific funding for this work.

**Competing interests:** The authors have declared that no competing interests exist.

most prevalent viruses detected in pig feces are PKoV, Mamastroviruses or Astrovirus 4 (PAstV), porcine Circovirus (PCV) and PSV [8, 9].

Kobuviruses belong to the *Picornaviridae* family. The genome is a single-stranded 8.2–8.3-kb RNA molecule that contains a large open reading frame coding for a polyprotein [10, 11]. Different species of kobuviruses have been found around the world in diverse animal species (pigs, cattle, sheep, goats, bats, rodents, felines, canines, etc.) and humans. It is suspected to be a pathogen that causes digestive disorders, particularly diarrhea in humans and animals, with transmission occurring via the fecal–oral route [12].

Porcine Sapelovirus (Family *Picornaviridae*, genus *Sapelovirus*) is a non-enveloped virus of 7.5–8.3 kb positive-polarity single-stranded RNA genome [13]. *Sapelovirus* genus is closely related to the genus Enterovirus and consists of three species: *Avian Sapelovirus*, *Sapelovirus A* (Porcine Sapelovirus [PSV]) and *Sapelovirus B* (simian Sapelovirus), with a single serotype [14]. PSV is transmitted via the fecal–oral route, and infection of pigs can be asymptomatic or associated with diarrhea, respiratory distress, encephalitis, skin lesions and reproductive tract disorders [15–17]. PSV is circulating in China, India, Korea, the United States, Brazil and Europe (Germany, the United Kingdom and Spain) [16–23]. Recently, PSV was detected and characterized for the first time in France, in Corsica. Importantly, the PSV-infected piglet from which the sequenced strain was isolated (PSV OPY-1-Corsica-2017; Genbank accession no. MH513612) was born and bred on the island of Corsica, suggesting local transmission [24].

Astroviruses are nonenveloped single-stranded RNA viruses with positive polarity, with an icosahedral capsid [25], that belong to the family *Astroviridae*, which includes two genera: *Mamastrovirus* (mammals) and *Avastrovirus* (avian) (ICTV, *Astroviridae*, 2019). Astroviruses can infect a large spectrum of animal species (pigs, deer, marine mammals, rodents, birds, pets, etc.) as well as humans [26]. Astrovirus infections are generally associated with more or less severe gastrointestinal signs in mammals (Mendez and Arias, 2007), but have also been detected in healthy individuals [27]. In humans, they cause intestinal disorders, particularly in children and immunocompromised individuals [28, 29].

The present study was conducted in Corsica, a French Mediterranean island, where livestock farming is a principal economic activity. In this region, more than 54,000 pigs, predominantly of the "Nustrale" breed, are bred using a traditional extensive farming system [30, 31]. Traditional extensive (or semi-extensive) outdoor system of pig farming is the main method of breeding. It favors contact with wild animals, which could result in sharing of pathogens such as hepatitis E virus (HEV) and Aujesky's disease agent [32, 33]. For HEV, we recently reported RNA detection in 9.2% of tested pig stool samples, with 75% of pig farms showing at least one positive sample [34]. Exploring the circulation of other enteric viruses in such pig farms could help to gain knowledge in the epidemiological cycle of HEV through the possible role of co-infection and super-infection. The main aim of this study was to detect and characterize three common porcine enteric viruses (PKoV, PAstV-1 and PSV) by molecular detection analysis of faeces collected within Corsican pigfarms.

## Materials and methods

### Study area, pig farms and sampling plan

Study area, samples/data collection, pig farms sampled, sampling plan and ethics statement are as described previously [34].

Briefly, **(i)** we collected fresh stool samples individually on the ground where pigs were pasturing and also intra-rectally using a glove with the help of a qualified technician (the individual level was control for each sample; feces were collected directly after defecation); **(ii)** three

types of breeding system operated in Corsica were included: seven outdoor extensive farms (E-farms), six outdoor semi-extensive farms (SE-farms) and three indoor closed farms (C-farms); **(iii)**. for each stool sample, the township, anonymous breeder code, breeding type, age and breed of pig and nature of the sample ("rectal" or "on the ground" feces) were recorded; **(iv)**. Four age categories were defined among the young pigs: 1–3 months, 3–4 months, 4–6 months and adults (older than 6 months). Samples from plots where post-weaning pigs were held together with older pigs (age mixed) were classified as the "Herd" group. Information on the individual health status of the pigs was not collected by a case report form. However, there is no apparent disease in the farms during the collect according to breeders. All the samples were collected between April and September 2017.

## RNA extraction and reverse transcription–quantitative polymerase chain reaction (RT–qPCR)

One gram of fecal sample was resuspended in 9 mL of phosphate-buffered saline and then centrifuged at 5,000 × g for 10 min. The resulting supernatant was collected and stored at −80˚C until processed. Viral RNA was extracted from 200 μL of supernatant using QIAamp Cador Pathogen on a QIAcube HT (Qiagen, Hilden, Germany) according to the manufacturer's instructions. Samples were spiked with an internal control (T4 and MS2 phages) before extraction, to monitor the extraction and subsequent steps, as described previously. Each pool was spiked before extraction with a predefined amount of MS2 bacteriophage in order to monitor the subsequent steps (nucleic acid purification, reverse transcription and PCR amplification) and to detect the presence of inhibitors and enzymatic reactions as described previously [35]. All extractions and RT-qPCRs were checked with the presence of a curve for phage and positive and negative controls respectively. Nucleic acids were eluted in 100 μL of RE buffer and stored at −80˚C.

Samples initially collected for the detection of HEV RNA were analyzed by RT–qPCR for PKoV, PSV and PAstV-1. Of the 919 samples initially collected, 908 were available for this study. Details of the three molecular assay [8, 36] are presented in Table 1 using an Applied Quant Studio 3 (Applied Biosystems, CA, USA). For PKoV RNA and PAstV-1 RNA detection, a RT–qPCR test was considered positive if negative controls were negative, positive controls were positive and an exponential curve was observed before a 35-Ct threshold.

For PSV RNA detection, the qPCR machine was programmed to perform a melt-curve analysis at the end of the run to ensure assay specificity; RT–qPCR results were considered positive if a melt curve was detected at between 83˚C and 85˚C and an amplification curve was observed before a 35-Ct threshold. The QuantiTect SYBR® Green PCR Kit was used for this biomolecular detection (Qiagen, Hilden, Germany).

**Table 1. RT–qPCR detection assays used in the study.**

| Viruses | Name of primers and probes | Sequences | References |
|---|---|---|---|
| PSV | FW: SYBR-PSV1 primer | GGCAGTAGCGTGGCGAGC | [36] |
| | REV: SYBR-PSV2 primer | CTACTCTCCTGTAACCAGT | |
| PKoV | FW: T-248-F-PKoV | TCTCTGACCTCTGAAGTGCACT | [8] |
| | REV: T-249-R-PKoV | TGAAGAAGCCATGTGTCTTGTC | |
| | Probe: T-250-PKoV-FAM | GGTTGCGTGGCTGGGAATCCAC | |
| PAstV-1 | FW: T217-F-PAstV-1 | CCAAAACCAGCAATCCGTCAA | |
| | REV: T218-R-PAstV-1 | GCCCCTAAAGCAACGATCGG | |
| | Probe: T-219-PAstV-1-VIC | TTCTTGTCAAGGATAATACGGGG | |

## Statistical analyses

The detection rate of RNA viruses (PKoV, PAstV-1 and PSV) in pig fecal samples was calculated at the individual level and the pig farm level. Positivity rate was also estimated in each subgroup, and a two-sided 95% confidence interval [95% CI] was calculated. Categorical variables were expressed as the number of cases (percentages). Frequencies were compared using the $\chi^2$ test or Fisher's exact test ($P < 0.05$). A bivariate analysis was carried out to identify the variables that were related to infection with each virus. The multivariate logistic regression analysis included variables that were related to outcome variables in the bivariate analysis with a $P$-value $< 0.2$ or a possible association. Odds ratios (ORs), including their 95% CIs, were calculated for the logistic regression models. As in Capai, F. [34], samples with no associated age (Herd group) were excluded from the multivariate analysis, and previous results for the detection rate of HEV RNA among pig feces were included in the analysis to estimate a possible association between coinfections with different viruses [37]. All statistical analyses were performed using the R program (http://www.r-project.org).

## Virus genome sequencing

Virus genome sequencing was performed for 26 stool samples as described previously [38]. Only, 26 analyses could be realized for financial reason and these 26 samples were randomly selected among the overall samples. A random RT–qPCR was performed using tagged random primers. A ProtoScript® II Reverse Transcriptase kit (New England Biolabs) was used for reverse transcription with random tagged primers, and Platinum® Taq High Fidelity polymerase enzyme (Thermo Fisher Scientific) with specific primers for amplification. After Qubit quantification using Qubit® dsDNA HS Assay Kit and Qubit 2.0 fluorometer (ThermoFisher Scientific), amplicons were fragmented (sonication) into fragments of 200 bp length. Libraries were built by adding barcodes for sample identification, and primers for amplification using the AB Library Builder System (ThermoFisher Scientific). To pool equimolar amounts of the barcoded samples, a quantification by quantitative PCR using an Ion Library TaqMan™ Quantitation Kit (Thermo Fisher Scientific) was performed. An automated Ion Chef instrument (ThermoFisher) was used for emulsion PCR of the pools and loading them on a 520 chip. Sequencing was performed using S5 Ion torrent technology (Thermo Fisher Scientific) following the manufacturer's instructions. Reads were trimmed (reads with quality score $< 0.99$ and length $< 100$ bp were removed, and the 30 first and 30 last nucleotides were removed from the reads), and de novo contigs were produced. These contigs were submitted to Blastn to determine the best reference sequences(s). A consensus sequence was obtained after mapping of the reads on the previously determined reference using CLC genomics workbench software 20.0.4 (Qiagen). The de novo contig was compared with the consensus sequence to ensure that the reference sequence did not affect the consensus sequence. Only sequences corresponding to enteric viruses found in pigs were selected for analysis.

## Results

As suggested by Arya, Antonisamy [39], we previously calculated the minimum sample size required to achieve the objectives related to the HEV (n = 176 stool specimens) [34]. Overall, we collected 919 pig feces samples from 16 pig farms selected according to location and breeding system.

For PSV RNA detection, using the SYBR green A range of temperatures around 84˚C (83–85˚C) was tolerated. Indeed, a one-nucleotide difference within the amplified sequence can impact the melt-curve dissociation temperature [40, 41].

## Viral RNA detection rate for the three viruses in feces from domestic pigs and univariate analysis

Overall, 908 samples were available for the detection of the three virus of interest, 310 were from E-farms, 396 from SE-farm and 201 from C-farm.

Our results showed viral RNA detection rates (i) of 62.0% [58.7–65.1] (n = 563/908) for PSV, (ii) of 44.8% [41.5–48.1] (n = 407/908) for PKoV and (iii) of 8.6% [6.8–10.6] (n = 78/908) for PAstV-1 (Table 2).

For PSV and PKoV, there was no statistical association with the type of breeding system; in contrast, PAstV-1 was detected more frequently in C-farms compared with SE- and E-farms (P-value = 9.6e–6) (Table 2).

Significant differences were observed for all three viruses according to age (P-value = 2.4e–13 for PAstV-1; 2.4e–12 for PKoV and 0.005 for PSV) (Table 2). RNA virus detection by age group showed a significant decrease in the rate of positive cases after three months for PAstV-1 (30.6% vs. 6.0%; P-value = 3.87e–7) and between 1 and 3 months (69.4%) and in adults (28.9%) for PKoV (P-value = 6.37e–12). For PSV, the detection rates by age group were between 61.0% and 77.5% (Fig 1). However, the positivity rate among pigs under six months of age was significantly lower than that in pigs older than 6 months (71.4% vs. 61.1%; P-value = 0.014; OR = 1.59, CI 95% 1.09–2.32).

## Description of coinfections in samples of pig feces

Table 3 lists all infections and coinfections detected in pig feces samples. Of the 908 samples tested, 697 samples were positive for at least one virus (76.8%). A total of 344 samples contained at least two distinct viral RNA (37.9%), of which 259 specimens (28.5%) were coinfected by two viruses, 78 specimens (8.6%) were coinfected by three viruses and seven specimens (0.8%) were positive for all four viruses (Table 3).

**Table 2. Viral RNA in stools of domestic pigs stratified by breeding system and age.**

| Factor | Condition | Number of samples | | positive (n) | PKoV RNA detection (%) | | P-value | positive (n) | PAstV-1 RNA detection (%) | | P-value | positive (n) | PSV RNA detection (%) | | P-value |
|---|---|---|---|---|---|---|---|---|---|---|---|---|---|---|---|
| | | N | % | | % | [95% CI] | | | % | [95% CI] | | | % | [95% CI] | |
| **Breeding systems** | E-farm | 310 | 34.1 | 136 | 43.9 | [38.3–49.6] | 0.91 | 17 | 5.5 | [38.3–49.6] | 9.6e–6 | 186 | 60.0 | [54.3–65.5] | 0.0094 |
| | SE-farm | 396 | 43.6 | 180 | 45.5 | [40.5–50.5] | | 27 | 6.8 | [4.5–9.8] | | 233 | 58.8 | [53.8–63.7] | |
| | C-farm | 201 | 22.1 | 91 | 45.3 | [38.2–52.4] | | 34 | 16.9 | [12–22.8] | | 143 | 71.1 | [64.3–77.3] | |
| **Age** | 1–3 months | 111 | 12.2 | 77 | 69.4 | [59.9–77.7] | 2.4e–12 | 34 | 30.6 | [22.2–40.1] | 2.4e–13 | 86 | 77.5 | [68.6–84.9] | 0.005 |
| | 3–4 months | 143 | 15.7 | 84 | 58.7 | [50.2–66.9] | | 9 | 6.3 | [2.9–11.6] | | 91 | 63.6 | [55.2–71.5] | |
| | 4–6 months | 162 | 17.8 | 71 | 43.8 | [36.0–51.8] | | 4 | 2.5 | [0.7–6.2] | | 120 | 74.1 | [66.6–80.6] | |
| | Adults (>6 months)* | 190 | 20.9 | 54 | 28.4 | [22.1–35.4] | | 17 | 8.9 | [5.3–13.9] | | 116 | 61.1 | [53.7–68.0] | |
| | Herds (age mixed) | 302 | 33.3 | 121 | 40.1 | [34.5–45.8] | | 14 | 4.6 | [2.6–7.7] | | 150 | 49.7 | [43.9–55.4] | |
| **All pigs** | | 908 | 100.0 | 407 | 44.8 | [41.5–48.1] | | 78 | 8.6 | [6.8–10.6] | | 563 | 62.0 | [58.7–65.1] | |

*6–8 months old (n = 47); Sow/Boar (n = 30); at least 6 months old (n = 108).

E-farm: Extensive farm; SE-farm: semi-extensive farm; C-farm: closed farm; Herds: pigs without associated age.

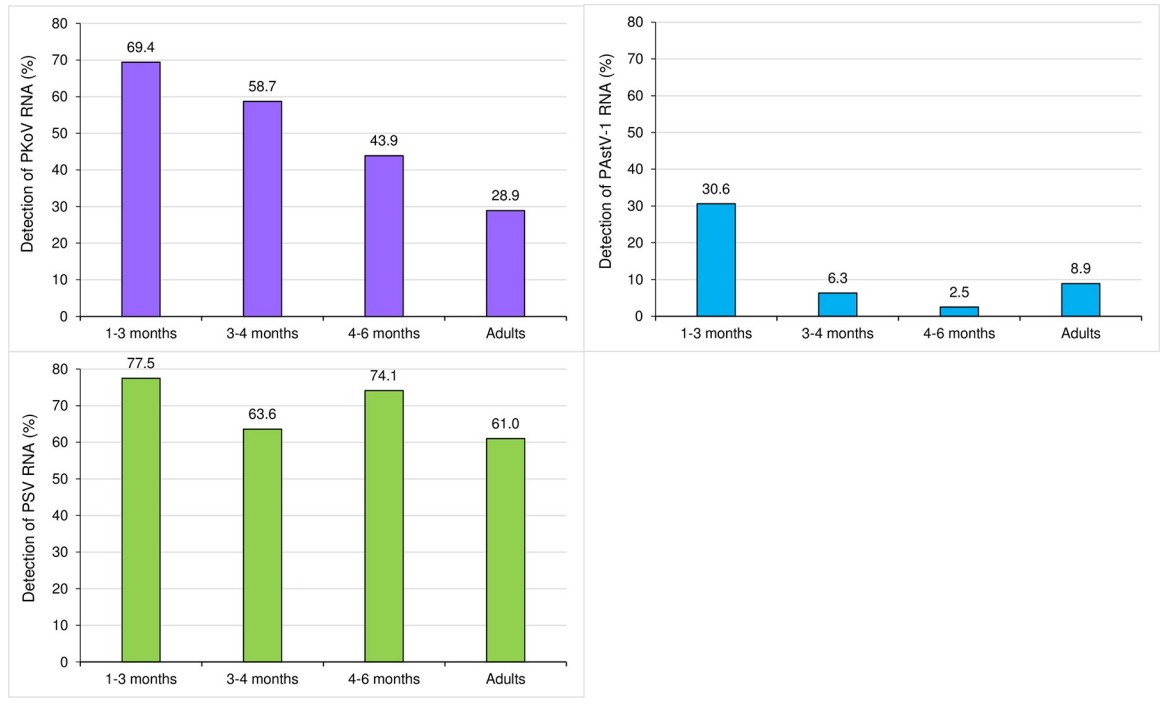

**Fig 1. Detection rate of each RNA virus by age group.**

**Detection rate of viral RNA at the farm level.** At least one pig was detected positive for PKoV RNA in each of the 16 pig farms sampled (100%), 62.5% for PAstV-1 infection (n = 10/16) and 93.8% for PSV infection (n = 15/16). The positivity rate at farm level ranged between 22.5% and 80.8%, 0.0% and 34.0%, and 0.0% and 87.5% for PKoV, PAstV-1 and PSV, respectively. The detail of the detection rates of the three viruses for each farm were presented on the S1 Table.

## Multivariate analysis: Associated factors identified for each viral infection

A multivariate logistic regression analysis was performed, and the results showed associations depending on the three viruses (Table 4). A strong association was observed between PKoV and PSV detection (OR = 3.36 [2.44–4.67]; $P$-value = 9.1e–15). PKoV was also associated with PAstV-1 codetection (OR = 2.16 [1.29–3.70]; $P$-value = 0.015) and young pigs under 4 months of age (OR = 3.11 [2.31–4.20]; $P$-value = 4.3e–10). PAstV-1 detection was also associated with

**Table 3. Coinfections in pig feces samples.**

| | N | | | | | | |
|---|---|---|---|---|---|---|---|
| **No infection** | 211 | **Number for each coinfection** | | | | | |
| **Single infection** | 353 | **PKoV** | **PSV** | **PAstV-1** | **HEV** | | |
| | | 102 | 229 | 9 | 13 | | |
| **Double infection** | 259 | **PKoV/PSV** | **PKoV/PAstV-1** | **PKoV/HEV** | **PSV/HEV** | **PAstV-1/PSV** | **PAstV-1/HEV** |
| | | 216 | 4 | 3 | 21 | 14 | 1 |
| **Triple infection** | 78 | **PKoV/PAstV-1/PSV** | **PKoV/HEV/PSV** | **PKoV/HEV/PAstV-1** | **HEV/PAstV-1/PSV** | | |
| | | 39 | 35 | 2 | 2 | | |
| **All four viruses** | 7 | | | | | | |

**Table 4. Statistical analysis of factors associated with each virus (multivariate logistic regression model with random effect at the farm level).**

| Viruses | Factor | Condition | OR | [95% CI] | P-value adjusted |
|---------|--------|-----------|-----|----------|------------------|
| PKoV | **Coinfection** | PSV | 3.36 | [2.44–4.67] | 9.1e−15 |
| | **Coinfection** | PAstV-1 | 2.16 | [1.29–3.70] | 0.015 |
| | **Age group** | 0–4 months | 3.11 | [2.31–4.20] | 4.3e−10 |
| PAstV-1 | **Coinfection** | PKoV | 2.25 | [1.32–3.90] | 0.014 |
| | **Age group** | >3 months | 0.18 | [0.11–0.29] | 2.8e−9 |
| PSV | **Coinfection** | HEV | 2.01 | [1.17–3.64] | 0.041 |
| | **Coinfection** | PKoV | 2.92 | [2.13–4.04] | 3.7e−8 |

Only factors with *P*-values < 0.05 are included.

age, with pigs over 3 months of age significantly less frequently infected than younger pigs (OR = 0.18 [0.11–0.29]; *P*-value = 2.8e–9). PSV infection was not associated with age or type of breeding.

**Next-generation sequencing (NGS).** In addition to the three viruses studied by RT–qPCR, NGS analysis was performed in a second phase on 26 random selected samples. Overall, consensus sequences corresponding to 10 different species of virus were obtained. PAstVs were detected in ten samples (38.5%), porcine stool/serum-associated circular virus in six (23.1%), Bocavirus in five (19.2%), Sapelovirus and Posavirus in four (15.4%), Circovirus in three (11.5%), Pasivirus in two (7.7%) and Rotavirus and porcine Enterovirus G in one (3.8%). Among these samples, 14 (54%) had consensus sequences corresponding to at least two different viruses.

Porcine astroviruses strains detected among our samples were closed to pig strains from different countries: Hungary (JQ340310 90.81–92% of identity); United States (KJ495987 93.36% of identity) and New Zealand (KJ495990 92.8% of identity). The main astroviruses found, were *Mammastrovirus* 3, *Porcine Astrovirus* 2 and 4. Concerning Bocaviruses strains detected, identity of 97.08% was determined with pigs from: a Hungarian domestic pig (KF206167), 95–97.65% with a Chinese pigs (KX017193; KM402139 and HM053693), 94.13–96.48% with an American pigs (KF025394 and KF025484) and 95.28% with a Croatian pig (KF206161). Concerning, the sequences obtained for the Sapelovirus, the strongest homologies were observed when compared with strain MH513612 also isolated in Corsica (92–99.91% of identity) [24].

For the other results, Table 5 summarizes all the sequences obtained, their length, the reference sequence to which each corresponds and the percentage of nucleotide identity with this sequence. Six of the recovered sequences had a quality score required for submission to Genbank and the accession numbers range from OL739527 to OL739532.

## Discussion

This study investigated the detection rate of three enteric viral infections, PKoV, Astrovirus-1 and Sapelovirus, in 908 pigs of different ages and breeding systems. To our knowledge, this work is the first to study factors associated with presence of viral RNA of PSV, PKoV and PAstV-1 in stools of domestic pigs in France.

### RNA detection rate of the three enteric viruses

PSV was the most prevalent (62%) and observed rates are higher than those described in the US, Brazil and India (7–31%) [9, 17, 20]. However, the existing results for PSV are not all

**Table 5. Diversity of sequences found in pig feces and percentage of identity with strains in the literature.**

| Samples | Accession number | Viruses | Sequence of reference | Accession number of the sequence of reference | Consensus length | % Cover | % Identity |
|---|---|---|---|---|---|---|---|
| 1 | / | / | / | / | / | / | / |
| 2 | OL739527 | Sapelovirus | Sapelovirus A strain OPY-1-Corsica-2017 polyprotein gene, complete cds | MH513612 | 5,927 | 96 | 92.46 |
| | / | Bocavirus | Porcine bocavirus strain G85-1AT-HU, complete genome | KF206167 | 3,507 | 100 | 97.08 |
| | / | Bocavirus | Porcine bocavirus 3C isolate pig/ZJD/China/2006, complete genome | JN681175 | 3,683 | 99 | 91.61 |
| | / | Porcine Astrovirus | Mamastrovirus 3 isolate PAstV_GER_L00855-K14_14–04_2014 genome assembly, complete genome: monopartite | LT898434 | 1,508 | 52 | 90.88 |
| 3 | / | Sapelovirus | Sapelovirus A strain OPY-1-Corsica-2017 polyprotein gene, complete cds | MH513612 | 492 | 100 | 93.09 |
| 4 | / | Bocavirus | Porcine bocavirus strain CH/HNZM, complete genome | KX017193 | 2,105 | 99 | 97.22 |
| | / | Porcine Astrovirus | Astrovirus wild boar/WBAstV-1/2011/HUN, complete genome | JQ340310 | 428 | 100 | 91.59 |
| | / | Porcine serum-associated circular virus | Porcine serum-associated circular virus isolate BR3, complete genome | KU203353 | 531 | 86 | 86.33 |
| | / | Enterovirus G | Enterovirus G isolate GER/F9-6/12-02-2013 polyprotein gene, partial cds | MF113376 | 956 | 100 | 86.72 |
| 5 | / | Sapelovirus | Porcine Sapelovirus isolate PSV_P1-3-3_Contig(g12h12) polyprotein gene, partial cds | KF705647 | 498 | 98 | 92.77 |
| | / | Sapelovirus | Sapelovirus A strain OPY-1-Corsica-2017 polyprotein gene, complete cds | MH513612 | 1,687 | 100 | 92.12 |
| | / | Sapelovirus | Sapelovirus A isolate HuN21 polyprotein gene, complete cds | MF440649 | 1,230 | 100 | 86.59 |
| | / | Sapelovirus | Sapelovirus A isolate PSV_GER_L00798-K11_14–02_2014 genome assembly, complete genome: monopartite | LT900497 | 1,894 | 100 | 87.08 |
| | / | Bocavirus | Porcine bocavirus strain 644-1DI-HR, complete genome | KF206161 | 4,580 | 100 | 95.28 |
| | / | Bocavirus | Porcine bocavirus isolate GD11, complete genome | KM402139 | 4,580 | 100 | 95.09 |
| | / | Porcine Astrovirus | Porcine astrovirus 2 clone KDC-6 ORF1ab gene, partial cds; and ORF2 gene, complete cds | KJ495987 | 466 | 100 | 93.36 |
| | / | Porcine Astrovirus | Porcine astrovirus 2 genes for ORF1ab, ORF1a, ORF2, complete cds, strain: PoAstV2/JPN/HgYa2-3/2015 | LC201588 | 422 | 100 | 87 |
| | / | Porcine Astrovirus | Astrovirus wild boar/WBAstV-1/2011/HUN, complete genome | JQ340310 | 533 | 100 | 90.81 |
| | / | Porcine stool-associated circular virus | Porcine stool-associated circular virus 7 isolate EP2-B, complete genome | KJ577813 | 1,191 | 99 | 82.59 |
| 6 | / | Posavirus | Posavirus 1 isolate PsaV_GER_L01017-K01_15–07_2015 genome assembly, complete genome: monopartite | LT898419 | 741 | 100 | 96.63 |
| | / | Posavirus | Posavirus sp. isolate 12144_61, complete genome | KX673217 | 559 | 100 | 97.5 |
| | / | Bocavirus | Porcine bocavirus 3 isolate IA159-3 NS1 and NP1 genes, complete cds; and VP1/VP2 gene, partial cds | KF025387 | 491 | 100 | 84.85 |
| | OL739528 | Porcine serum-associated circular virus | Porcine serum-associated circular virus isolate BR3, complete genome | KU203353 | 929 | 99 | 86.03 |
| 7 | / | / | / | / | / | / | / |
| 8 | / | / | / | / | / | / | / |

(*Continued*)

**Table 5.** (Continued)

| Samples | Accession number | Viruses | Sequence of reference | Accession number of the sequence of reference | Consensus length | % Cover | % Identity |
|---------|------------------|---------|-----------------------|-----------------------------------------------|------------------|---------|------------|
| 9 | OL739529 | Circovirus | Circovirus sp. isolate PoCirV_VIRES_JL01_C5 capsid protein gene, partial cds | MK377643 | 1,038 | 100 | 85.17 |
| | / | Circovirus | Circovirus sp. isolate PoCirV_VIRES_GX05_C4 replicase gene, partial cds | MK377558 | 639 | 64 | 89.54 |
| | / | Porcine Astrovirus | Porcine astrovirus 2 genes for ORF1ab, ORF1a, ORF2, complete cds, strain: PoAstV2/JPN/Ishi-Ya4/2015 | LC201589 | 879 | 78 | 82.61 |
| | OL739530 | Porcine Astrovirus | Porcine astrovirus 4 genes for ORF1ab, ORF1a, ORF2, complete cds, strain: PoAstV4/JPN/Bu5-10-2/2014 | LC201603 | 989 | 100 | 83.64 |
| 10 | / | Bocavirus | Porcine bocavirus 1 pig/ZJD/China/2006, complete genome | HM053693 | 383 | 100 | 97.65 |
| 11 | / | Porcine Astrovirus | Porcine astrovirus 4 isolate 15–12, complete genome | KU764486 | 1,624 | 99 | 87.98 |
| | / | Porcine Astrovirus | Porcine astrovirus 4 isolate 15–13, complete genome | KU764484 | 1,624 | 99 | 88.21 |
| | / | Porcine stool-associated circular virus | Porcine stool-associated circular virus 7 isolate EP3-C, complete genome | KJ577814 | 719 | 100 | 86.77 |
| 12 | / | Posavirus | Posavirus sp. isolate 17668_12, complete genome | KX673279 | 1,606 | 93 | 85.53 |
| 13 | / | Porcine Astrovirus | Porcine astrovirus 4 strain JXJA, complete genome | KX060808 | 1,187 | 100 | 89.9 |
| | / | Bocavirus | Porcine bocavirus 3 isolate IL330 NS1 and NP1 genes, complete cds; and VP1/VP2 gene, partial cds | KF025394 | 483 | 100 | 96.48 |
| 14 | / | Porcine Astrovirus | Porcine astrovirus 4 strain 35/USA, complete genome | JF713713 | 1,798 | 100 | 90.12 |
| | / | Porcine Astrovirus | Porcine astrovirus 4 strain CH/JXZS/2014, complete genome | KX060809 | 1813 | 100 | 90.13 |
| 15 | OL739531 | Porcine Astrovirus | Mamastrovirus 3 isolate PoAstV_VIRES_HeB02_C4 ORF1ab and ORF1a genes, partial cds | MK378508 | 2,192 | 100 | 90.81 |
| | / | Circovirus | Circovirus sp. isolate PoCirV_VIRES_SD02_C2 replicase gene, partial cds | MK377699 | 557 | 100 | 88.6 |
| 16 | OL739532 | Circovirus | Circovirus sp. isolate PoCirV_VIRES_HeB04_C3 capsid protein gene, partial cds | MK377607 | 777 | 100 | 90.27 |
| 17 | / | Porcine Astrovirus | Porcine astrovirus 4 genes for ORF1ab, ORF1a, ORF2, complete cds, strain: PoAstV/JPN/MoI2-1-2/2015 | LC201610 | 869 | 100 | 89.31 |
| | / | Porcine Astrovirus | Porcine astrovirus 2 clone NZP-93_Subtype_2 ORF1ab gene, partial cds | KJ495990 | 375 | 100 | 92.8 |
| | / | Bocavirus | Porcine bocavirus 3 isolate IA13-1 VP1/VP2 gene, complete cds | KF025484 | 931 | 100 | 94.13 |
| | / | Porcine Astrovirus | Mamastrovirus 2 isolate U083, complete genome | KY940077 | 2,043 | 100 | 80.33 |
| | / | Porcine Astrovirus | Mamastrovirus 3 isolate PoAstV_VIRES_GZ04_C10 ORF1ab and ORF1a genes, partial cds | MK378502 | 869 | 100 | 89.77 |
| 18 | / | Porcine Astrovirus | Porcine astrovirus 4 genes for ORF1ab, ORF1a, ORF2, complete cds, strain: PoAstV4/JPN/Ishi-Ya7-1/2015 | LC201613 | 1,970 | 91 | 89.22 |
| | / | Porcine Astrovirus | Mamastrovirus 2 isolate U083, complete genome | KY940077 | 5,230 | 85 | 87.06 |
| | / | Porcine Astrovirus | Mamastrovirus 3 isolate PAstV_GER_L00855-K14_14–04_2014 genome assembly, complete genome: monopartite | LT898434 | 4,050 | 100 | 85.87 |
| 19 | / | / | / | / | / | / | / |
| 20 | / | Sapelovirus | Sapelovirus A strain OPY-1-Corsica-2017 polyprotein gene, complete cds | MH513612 | 2,360 | 100 | 99.91 |
| | / | Pasivirus | Swine pasivirus SPaV1/US/17-50816IA60467-1/2001 polyprotein gene, complete cds | MG674090 | 1,186 | 100 | 77.88 |
| | / | Porcine stool-associated circular virus | Porcine serum-associated circular virus isolate BR2, complete genome | KU203352 | 1,168 | 99 | 92.76 |

(*Continued*)

**Table 5.** (Continued)

| Samples | Accession number | Viruses | Sequence of reference | Accession number of the sequence of reference | Consensus length | % Cover | % Identity |
|---|---|---|---|---|---|---|---|
| **21** | / | / | / | / | / | / | / |
| **22** | / | Rotavirus | Porcine rotavirus B isolate PoRVB_VP2_VIRES_NM01_C2 VP2 gene, partial cds | MK379346 | 55 | 100 | 98.18 |
| **23** | / | / | / | / | / | / | / |
| **24** | / | Posavirus | Posavirus sp. isolate 17668_12, complete genome | KX673279 | 3,003 | 100 | 88.71 |
| | / | Pasivirus | Pasivirus A isolate SPaV-A GER L01061-K07 15–03 2015 genome assembly, complete genome: monopartite | LT898422 | 451 | 100 | 83.44 |
| | / | Porcine stool-associated circular virus | Porcine stool-associated circular virus 7 isolate EP3-C, complete genome | KJ577814 | 702 | 100 | 83.25 |
| **25** | / | / | / | / | / | / | / |
| **26** | / | Posavirus | Posavirus sp. isolate 17668_13_2, complete genome | KX673281 | 1,282 | 99 | 89.31 |
| | / | Posavirus | Posavirus 3 strain 10611 polyprotein gene, complete cds | KT833079 | 860 | 100 | 94.07 |
| | / | Posavirus | Posavirus strain 7048 polyprotein gene, partial cds | KT833076 | 1.121 | 99 | 87.92 |

performed with the same method (SYBR, classical RT-qPCR, NGS) so it is complicated to compare the results in the literature.

PKoV RNA was detected in almost half of samples, which is similar to data reported from pig farms in China (45.7%) [42] and Japan (45.4%) [43]. In five European countries, PKoV infections have been described previously as highly prevalent in both diarrheic and healthy pigs, 54.5% and 58.2%, respectively [8].

PAstV-1 RNA was detected in less than 10% of our samples, but comparison is difficult because other studies were performed at the genus level (*Mamastrovirus*: 52% positive) [9] or for all PAstVs combined (Thailand: 6.5%; MN, USA: 62%; Slovakia: 93.2%) [44–46].

## Very early exposure of pigs

The analysis of the detection of viral RNA according to pig age showed that for all viruses tested, pigs < 4 months old were consistently the age group exhibiting the highest rate. This association between age and rate of infection was confirmed for PKoV and PAstV-1 in the multivariate analysis. It may correlate with high exposure after weaning (about 2 months after birth) but also suggest persisting presence of viruses in the breeding environment and loss of maternal immunity. Piglet passive immunity is derived from colostrum and not from breastfeeding [47], with a decrease in immunoglobulins A, G and M after farrowing [48]. This trend was also observed during our previous study of HEV) [34]. Correlation of decreasing positivity rates and increasing age of pigs is described for Kobuviruses in Italy [49] and East Africa [50]. A higher PKoV detection rate in young piglets has also been reported in other studies [10, 51, 52]. However, the kinetics of infection were different in each study and may depend on the organization of each farm and other environmental characteristics. Study in different age-group 0-1months, 1-2months, etc. or follow-up of individual animals could help confirming this hypothesis.

In traditional Corsican farms, the average slaughter age is higher (12–18 months) compared with industrial farms. Therefore, at the time of slaughtering the majority of *Nustrale* Corsican pigs will have cleared replicative viral infection. The same finding was already reported for hepatitis E virus [34]. In contrast, the situation is different for Sapeloviruses with rates of replicating infection at 60% in adult pigs. Farming type does not seem to influence the detection rates of studied viruses but still need confirmation from future studies.

## A potentially very broad spectrum of viruses in pig feces

In this study, 37.9% of pigs were coinfected with at least two different viruses (including HEV). Strong trends were observed in the multivariate analysis regarding the associations between different viral infections (PKoV/PSV, PKoV/PAstV-1, PSV/HEV and PSV/PKoV). This wide variety of viruses in the feces was confirmed with the NGS method, which showed the presence of other viruses frequently found in pigs: Posavirus, other Astroviruses, Bocavirus, Enterovirus G, Circovirus, Pasivirus and Rotavirus. Coinfections were also evidenced in the NGS results, with more than half of the samples containing consensus sequences corresponding to at least two different viruses. These results are in line with previous studies reporting multiple coinfections of farmed pigs with porcine enteric viruses [6, 37], especially in pigs with diarrhea [53–55]. Our results are also interesting concerning the epidemiological cycle of HEV because co-infections can diminish the immune system of pigs, change the symptomatology of infections, [56, 57]. For example, experimental co-infection of HEV with Porcine Reproductive and Respiratory Syndrome Virus significantly prolonged HEV shedding [58].

Our study has several limitations. First, it was not initially designed to study these three viruses but to determine the prevalence of the HEV in pig farms in Corsica. Differences in detection could exist depending on criteria such as sampling time and place, age of pigs and clinical background of the tested animal population (diarrheic or healthy pigs). The lack of collection of clinical information (diarrhea or other symptoms) for the pigs included may have led to a bias in the analysis of the data; these data could have been essential for the improvement of knowledge concerning the studied viruses. The number of viruses studied could also have been larger to better assess the presence of major enteric viruses in Corsican pig farms. Finally, the lack of phylogenetic analysis of the different strains found is also a major limitation of our study. Indeed, the sequences obtained being in very varied zones of the genome of the various viruses, the phylogenetic analyses could not be realized.

In future studies, information about possible symptoms in pigs should be collected. It would also be useful to evaluate the phylogeny of the different strains found and to set up an RT–qPCR primer system to distinguish the different strains of Sapelovirus. Moreover, in view of the large variety of viruses present in the pig feces and the availability of microfluidic PCR technology, it would be helpful to set up microarrays that can detect all the principal known porcine enteric viruses. Moreover, the real impact of Kobuvirus, Astroviruses, Sapelovirus and other enteric viruses on animal health and breeding systems remains largely unknown and needs further epidemiological studies.

In conclusion, this study provides first insight on the presence of three common porcine enteric viruses in France and showed that they are frequently encountered in Corsica in pig farms using the traditional extensive breeding. The three viruses studied were found on almost all the farms, indicating widespread distribution. Moreover, the pigs tested were born and bred in Corsica, which demonstrates endemic local circulation. Whether such infections and co-infections can affect the productivity, impact the growth of pigs or cause immune weakness remains to be established. So far, this study has to be considered as a first step in the study of enteric viruses in Corsican pig farms.

## Supporting information

**S1 Table. Positivity rates for the three viruses by pig farm.**
(DOCX)

**S2 Table.**
(DOCX)

## Acknowledgments

The authors are grateful to all breeders for their participation in the study.

### Ethical approval

The main part of the molecular analysis was carried out on fresh stools from the ground without contact with animals (98.4% of the samples). Fifteen of the 908 stools collected were intra-rectal samples, and were initially performed in a diagnostic setting (diarrheic pigs). We took an aliquot of these samples which have been collected by a qualified technician in the presence of the breeder. In this case, the study was exempt from ethical authorization because we did not handle animals and the intra-rectal samples were performed independently of the study with a diagnostic aim. All the breeders gave their explicit agreement for the collection of the samples, and all the data were anonymized.

## Author Contributions

**Conceptualization:** Lisandru Capai, Oscar Maestrini, François Casabianca, Xavier de Lamballerie, Rémi N. Charrel, Alessandra Falchi.

**Data curation:** Lisandru Capai.

**Formal analysis:** Géraldine Piorkowski, Shirley Masse.

**Funding acquisition:** Alessandra Falchi.

**Investigation:** Lisandru Capai, Oscar Maestrini.

**Methodology:** Lisandru Capai, Géraldine Piorkowski, Shirley Masse, Rémi N. Charrel.

**Supervision:** Rémi N. Charrel, Alessandra Falchi.

**Writing – original draft:** Lisandru Capai, Alessandra Falchi.

**Writing – review & editing:** François Casabianca, Xavier de Lamballerie, Rémi N. Charrel, Alessandra Falchi.

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
