## [Decision Letter · Decision Letter 0]

11 Sep 2021

PONE-D-21-15693Detection of porcine enteric viruses (Kobuvirus, Mamastrovirus and Sapelovirus) in domestic pigs in Corsica, France.PLOS ONE

Dear Dr. Capai,

Thank you for submitting your manuscript to PLOS ONE. After careful consideration, we feel that it has merit but does not fully meet PLOS ONE’s publication criteria as it currently stands. Therefore, we invite you to submit a revised version of the manuscript that addresses the points raised during the review process. See the reviewers' comments on your manuscript.

The work is well designed, written and the results are interesting; however, there are several important points of concern that need to be addressed.

Sincerely

We look forward to receiving your revised manuscript.

Kind regards,

Humberto Rodney Colina Muñoz, PhD

Academic Editor

PLOS ONE

Journal Requirements:

“The authors are grateful to all breeders for their participation in the study. Part of this study was funded by the territorial collectivity of Corsica (CDC) and the National Institute of Health and Medical Research (INSERM). The Hepatitis E Virus project of UR BIOSCOPE was also financed by the CDC.”

“Part of this study was funded by the territorial collectivity of Corsica (CDC) and the National Institute of Health and Medical Research (INSERM). The Hepatitis E Virus project of UR BIOSCOPE was also financed by the CDC.”

3. "In your Data Availability statement, you have not specified where the minimal data set underlying the results described in your manuscript can be found. PLOS defines a study's minimal data set as the underlying data used to reach the conclusions drawn in the manuscript and any additional data required to replicate the reported study findings in their entirety. All PLOS journals require that the minimal data set be made fully available. For more information about our data policy, please see http://journals.plos.org/plosone/s/data-availability.

5. Please include a caption for figure 1.

Reviewers' comments:

Reviewer's Responses to Questions

**Comments to the Author**

1. Is the manuscript technically sound, and do the data support the conclusions?

Reviewer #1: Yes

Reviewer #2: Yes

2. Has the statistical analysis been performed appropriately and rigorously? 

Reviewer #1: Yes

Reviewer #2: Yes

3. Have the authors made all data underlying the findings in their manuscript fully available?

Reviewer #1: No

Reviewer #2: Yes

4. Is the manuscript presented in an intelligible fashion and written in standard English?

Reviewer #1: Yes

Reviewer #2: Yes

5. Review Comments to the Author

Reviewer #1: This is a nice, descriptive work, aimed to study the frequency of detection of 3 porcine, enteric viruses in herds form Corsica, previously investigated for HEV infection. The study is of interest to the field given that these are very poorly studied pathogens, and thus very little is known concerning their epidemiology and distribution.

The research was properly designed, the draft is well written, and the results are satisfactorily presented in general. However I have several major concerns that must be addressed.

Major points

1. Authors should re define the main objective of the study. Should clarify what is " to investigate the circulation....". The objective should not be a mere statement, but must be descriptive.

2. Sampling:

a. Were the animals healthy? Please clarify this in Mat and Meth section.

b. How di you collect stool samples' individually' from te ground? Can you be sure that that feces you collected corresponded to only 1 animal?

c. Related to b. Did the pigs (E or SE-farm) share breeding area with other animals? Including occasional feral mammals? Did you investigate presence of feces coming from animals, others than pigs?

c. Th amount of samples per breeding system should be included also in the text.

3. How were the 26 samples selected for NGS analysis? Which was the criterion? Please clarify.

4. Two phages was used as QC of the extraction process. However, it is not clear which was the % of recovery after processing the samples. Please include this information.

5. It is quite surprising that the younger animals (1-2 months), supposed to still have maternally-derived antibodies, presented the highest frequency rate for all viruses. I would expected that for animals > 3 or 4 months, after loosing those protective antibodies (as seen,i.g, for HEV). Maybe vertical transmission is not negligible in the enteric viruses investigated in this work, and it indeed may be playing a role in a scenario of heavy endemic viral transmission. Please provide a deeper discussion in this issue.

6. Related to point 5, and considering some technical issues, please provide a summary (mean, SD, etc) of Ct values for the samples, particularly for the SYBR green detection approach. Did the authors performed end-point PCR approaches to confirm the findings? Please discuss the first point.

7. Authors must perform a preliminary phylogenetic reconstruction for PAstV sequences, at least using ORF2. Please submit and provide Genbank accession numbers for the employed sequences.

8. Delete lines 260-263.

Minor points

Line 150: Check spelling

Line 241: Percentage of homology is not correct. Substitute by: percentage of nucleotide identity. Same for 231-232, etc.

Table 4: No of Pig Farm

Color should be avoided in the tables.

Thank you.

Reviewer #2: Capai et al present a study investigating the presence of three viruses (Porcine Sapelovirus, Porcine Kobuvirus and

porcine Astrovirus in 908 fresh pig stool samples using differents molecular assays. Additionally, they investigated the presence of enteric viruses in 26 samples using NGS.

Although in many cases, these viruses have been detected in animals without clinical disease, the impact of these viruses on pig disease is not well understood and consequently the information generated may be of interest in this field.

The first part of this work is well designed and represents the main results that indicate a high circulation of the viruses studied and its relationship with the presence of other viruses and the age group. However, I find some inconsistencies in the second part of this study. Data from the analysis by NGS of 26 samples are added and it is not explained which was the selection criteria of these samples or the relationship with the previous study. Furthermore, since a phylogenetic analysis was not possible, the information provided cannot be considered representative of the strains detected in this study. Consequently, I consider that this manuscript should be revised to improve its value.

Other minor comments

1-Line 88-92: in these lines it is proposed to relate the epidemiological cycle of HEV with coinfectant viruses. However, no mention is made of the results obtained in the discussion.

2- Line 96: The time period in which the samples were collected is not described. It should be clarified in the manuscript.

3- Line 243. Table 4. I find this table unnecessary in the manuscript. It could be placed as supplementary material. I do not find in it that the importance of these results is discussed.

4- Line 258: What is the influence of using Syber Green PCR for the detection of PSV? RT-qPCR was used for the other 2 viruses. Can the use of this method explain the differences in prevalences found by other researchers?

5- Line 329: How did you demonstrate the endemic circulation of these viruses?

6. PLOS authors have the option to publish the peer review history of their article (what does this mean?). If published, this will include your full peer review and any attached files.

Reviewer #1: No

Reviewer #2: No

---

## [Author Response · Author response to Decision Letter 0]

1 Nov 2021

Dear Reviewers,

We would like to thank you for the constructive and helpful comments to our manuscript. 

We have revised the manuscript and submitted a revised manuscript using the “Track Changes” tool in Microsoft Word for your consideration. 

Please find below our point-by-point response.

All co-authors have read and approved the manuscript.

Should you have any further questions please do not hesitate to contact us.

Kind regards,

Lisandru Capai.

Reviewer(s)' Comments to Author:

• Reviewer #1:

“This is a nice, descriptive work, aimed to study the frequency of detection of 3 porcine, enteric viruses in herds form Corsica, previously investigated for HEV infection. The study is of interest to the field given that these are very poorly studied pathogens, and thus very little is known concerning their epidemiology and distribution.

The research was properly designed, the draft is well written, and the results are satisfactorily presented in general. However, I have several major concerns that must be addressed.”

Major points

1. Authors should re define the main objective of the study. Should clarify what is " to investigate the circulation....". The objective should not be a mere statement, but must be descriptive.

Author’s response: We changed the sentence “The main aim of this study was to detect and characterize three common porcine enteric viruses (PKoV, PAstV-1 and PSV) by molecular detection analysis of faeces collected within Corsican pigfarms.” Line 92 to 94. 

2. Sampling:

a. Were the animals healthy? Please clarify this in Mat and Meth section.

Author’s response: We added a sentence on the Mat and Meth section: “Information on the individual health status of the pigs was not collected by a case report form. However, there is no apparent disease in the farms during the collect.” Line 107 to 110.

b. How did you collect stool samples' individually' from the ground? Can you be sure that that feces you collected corresponded to only 1 animal?

Author’s response: We have collected individually from the ground (the individual level was control for each sample; feces were collected directly after defecation). Precision added on the Mat and met part Line 101.

c. Related to b. Did the pigs (E or SE-farm) share breeding area with other animals? Including occasional feral mammals? Did you investigate presence of feces coming from animals, others than pigs?

Author’s response: Indeed, in extensive or semi-extensive farms pigs can share the area of breeding with other wild/domestics animals (like wild boars, cows, or sheep) that get through the fences (SE) or that the pigs cross during their routes (E). However, in this study we have only collected pig samples.

d. The amount of samples per breeding system should be included also in the text.

Author’s response: We added a sentence on the Results part “Overall, 908 samples were available for the detection of the three virus of interest, 310 were from E-farms, 396 from SE-farm and 201 from C-farm.” Line 187 to 188.

3. How were the 26 samples selected for NGS analysis? Which was the criterion? Please clarify.

Author’s response: Only, 26 analyses could be realized for financial reason and these 26 samples were randomly selected among the overall samples. We added a sentence in the material and methods part. Line 153-154

4. Two phages was used as QC of the extraction process. However, it is not clear which was the % of recovery after processing the samples. Please include this information.

Author’s response: We added a sentence on the Mat et Methods part “Each pool was spiked before extraction with a predefined amount of MS2 bacteriophage in order to monitor the subsequent steps (nucleic acid purification, reverse transcription and PCR amplification) and to detect the presence of inhibitors and enzymatic reactions as described. All extractions and RT-qPCRs were checked with the presence of a curve for phage and positive and negative controls respectively.” Line 118 to 123.

5. It is quite surprising that the younger animals (1-2 months), supposed to still have maternally-derived antibodies, presented the highest frequency rate for all viruses. I would expected that for animals > 3 or 4 months, after loosing those protective antibodies (as seen,i.g, for HEV). Maybe vertical transmission is not negligible in the enteric viruses investigated in this work, and it indeed may be playing a role in a scenario of heavy endemic viral transmission. Please provide a deeper discussion in this issue.

Author’s response: This point confirms the hypotheses that passive immunity of piglets is due to colostrum and not to breastfeeding. Furthermore, we believe that it is possible that the highest percentages of infections occur when the piglets leave the mother and are more strongly exposed to viruses in their new environment. In Corsica, these occurs 40 and 60 days after the birth. Study in different age-group 0-1months, 1-2months, etc. or follow-up of individual animals could help answering this question.

6. Related to point 5, and considering some technical issues, please provide a summary (mean, SD, etc) of Ct values for the samples, particularly for the SYBR green detection approach. Did the authors performed end-point PCR approaches to confirm the findings? Please discuss the first point.

Author’s response: We added a Summary table concerning the results of qRT-PCR for the three viruses of interest in the supplemental data (Supplementary Table 2). All our results were checked with classical positive and negative controls. 

7. Authors must perform a preliminary phylogenetic reconstruction for PAstV sequences, at least using ORF2. Please submit and provide Genbank accession numbers for the employed sequences.

Author’s response: The different sequences obtained cannot be analyzed in phylogeny because they do not concern the same areas of the genome. Even for PAstV sequences it is not possible to do phylogeny on ORF2 because they are on other regions of the genome and the alignment is not feasible. We will be attentive to this in our next studies in order to deepen the results with phylogeny. We have submitted the sequences and are waiting for the accessions numbers. You will find the proof of submission attached. The accession numbers of our sequences will be add in the Table 5.

8. Delete lines 260-263.

Author’s response: We deleted these lines.

Minor points

Line 150: Check spelling � Done

Line 241: Percentage of homology is not correct. Substitute by: percentage of nucleotide identity. Same for 231-232, etc. � Done

Table 4: No of Pig Farm � Done

Color should be avoided in the tables. � Done

Thank you.

• Reviewer #2:

Capai et al present a study investigating the presence of three viruses (Porcine Sapelovirus, Porcine Kobuvirus and porcine Astrovirus in 908 fresh pig stool samples using differents molecular assays. Additionally, they investigated the presence of enteric viruses in 26 samples using NGS. 

Although in many cases, these viruses have been detected in animals without clinical disease, the impact of these viruses on pig disease is not well understood and consequently the information generated may be of interest in this field. The first part of this work is well designed and represents the main results that indicate a high circulation of the viruses studied and its relationship with the presence of other viruses and the age group.

Author’s response: We thanked the reviewer for his comment.

However, I find some inconsistencies in the second part of this study. Data from the analysis by NGS of 26 samples are added and it is not explained which was the selection criteria of these samples or the relationship with the previous study. 

Author’s response: Only, 26 analyses could be realized for financial reason and these 26 samples were randomly selected among the overall samples. We added a sentence in the material and methods part. Line 153-154

Furthermore, since a phylogenetic analysis was not possible, the information provided cannot be considered representative of the strains detected in this study. Consequently, I consider that this manuscript should be revised to improve its value.

Author’s response: We have submitted the sequences and are waiting for the accessions numbers. You will find the proof of submission attached. The accession numbers of our sequences will be add in the Table 5.

Other minor comments

1-Line 88-92: in these lines it is proposed to relate the epidemiological cycle of HEV with coinfectant viruses. However, no mention is made of the results obtained in the discussion.

Author’s response: We added sentences in the discussion part Line 314-318.

2- Line 96: The time period in which the samples were collected is not described. It should be clarified in the manuscript.

Author’s response: We had a sentence on the Mat et Methods part “All the samples were collected between April and September 2017.” Line 110-111.

3- Line 243. Table 4. I find this table unnecessary in the manuscript. It could be placed as supplementary material. I do not find in it that the importance of these results is discussed.

Author’s response: We transferred this table on the Supplementary material.

4- Line 258: What is the influence of using Syber Green PCR for the detection of PSV? RT-qPCR was used for the other 2 viruses. Can the use of this method explain the differences in prevalences found by other researchers?

Author’s response: Indeed, the different method used to detect the virus make the results not comparable. We added a sentence to precise this point. Line 272-274

5- Line 329: How did you demonstrate the endemic circulation of these viruses?

Author’s response: In our opinion, the presence of the different viruses in numerous farms across the island and the observed prevalences may lead to the hypothesis of a real endemic circulation on the island.

De : gb-admin@ncbi.nlm.nih.gov [mailto:gb-admin@ncbi.nlm.nih.gov] 

Envoyé : mercredi 27 octobre 2021 17:43

À : Lisandru CAPAI <capai_l@univ-corse.fr>; lisandru.capai@gmail.com

Objet : GenBank Submissions grp 8269814

Dear Dr. Capai:

We have received the following 55 sequence submission(s) from you:

BankIt2512508 : (55)

It appears that these may be metagenomically derived virus sequences.

Please clarify whether you:

[a] Purified viral particles and sequenced the DNA.

[b] Sequenced mixed DNA from a metagenomic source and then

binned the viral sequences into assemblies.

[c] assembled third party reads obtained from a public database.

Please explain exactly how these sequences were generated.

Send your reply to: gb-admin@ncbi.nlm.nih.gov

If we do not hear from you by Nov 10, 2021, all of your submission(s) 

will be deleted from the processing queue.

Thank you for your attention. We will not assign GenBank Accession 

Numbers or further process your submissions until we hear from you. 

For your reference, please find your preliminary records below.

Please reply using the current Subject line.

Sincerely,

Linda Frisse, PhD 

The GenBank Submissions Staff

Bethesda, Maryland USA

gb-admin@ncbi.nlm.nih.gov (for replies and updates to current records)

info@ncbi.nlm.nih.gov (for general questions regarding GenBank)

preliminary GenBank Flatfile(s):

LOCUS Seq01 5927 bp RNA linear VRL 26-OCT-2021

DEFINITION Sapelovirus A strain PSV-1-LC-Corsica-2017.

ACCESSION 

VERSION

KEYWORDS .

SOURCE Sapelovirus A

 ORGANISM Sapelovirus A

 Viruses; Riboviria; Orthornavirae; Pisuviricota; Pisoniviricetes;

 Picornavirales; Picornaviridae; Sapelovirus.

REFERENCE 1 (bases 1 to 5927)

 AUTHORS Capai,L., Piorkowski,G., Maestrini,O., Casabianca,F., de

 Lambellerie,X., Charrel,R. and Falchi,A.

 TITLE Direct Submission

 JOURNAL Submitted (26-OCT-2021) URBIOSCOPE 7310, Universite de Corse

 Pasquale Paoli, CAMPUS GRIMALDI BAT PPDB, CORTE, CORSICA 20250,

 FRANCE

COMMENT Bankit Comment: LocalID:Seq01.

 ##Assembly-Data-START##

 Assembly Method :: CLC GENOMICS v. WORKBECH 21.0.5

 Sequencing Technology :: IonTorrent

 ##Assembly-Data-END##

FEATURES Location/Qualifiers

 source 1..5927

 /organism="Sapelovirus A"

 /mol_type="genomic RNA"

 /isolate="Corsica, France"

 /isolation_source="Pig feces sample"

 /host="Domestic pig"

 /db_xref="taxon:686984"

 /country="France"

 /collection_date="2017"

BASE COUNT 1809 a 1165 c 1326 g 1627 t

ORIGIN 

 1 ccttaaggtg gttgtatcca cataccccac cctcccttcc aaagcggatg gacaaacgga

 61 ctttgactta tggcgagttt acacggtatg atttttggat acacttgaat ggtagtagcg

 121 tggcgagcta tggaaaaatc gcaattgtcg atagccatgt tagtgacgcg cttaggcgtg

 181 ctcctttggt gattcggcga ctggttacag gagagtaggc agtgagctat gggcaaacct

 241 ctacagtatt acttagaggg aatgtgcaat tgagacttga cgagcgtctc tttgagatgt

 301 ggcgcatgct cttggcatta ccatagtgag cttccaggtt gggagacctg gactgggcct

 361 atactacctg atagggtcgc ggctggccgc ctgtaactag tatagtcagt tgaaaccccc

 421 ccatggaatc tactactact ctttcatttt gcaactggat ccctaagaag cagagagccc

 481 gtgtgtacct caccaccagt gtgacacacg agaaatcaat tgggccatat acgtatgtgg

 541 tctcagacat gatcatgaaa gaaaatagta gaacctctct tgccatggca tacgtggaag

 601 ggaagacact agtgttcaac actggaactc aattgggtca agtacattca gccaacactg

 661 gaaacaaacc acaaggtgca tacaatcatg gttctggcag tataacacag gttaactact

 721 atggctctga ctactcacag gcatggaatc ccacacaaca acagatggac ccatcgcaat

 781 tcaccaaacc cgtcacagaa attgctagta tggtagcagg gtccaacaca ccagcaggac

 841 ctgcactcaa ggcacctgac aaggaagaag aaggttacag tgacagactg atgcagctaa

 901 cagctggcaa ctcctgtata acaacacagg aggcggcaaa ggcagtggtg gcatatggcc

 961 aatggcctag ctataacata gatggaggag aacatttgga cctggccacc actcctggaa

 1021 cagcagtaga caggttctat acttttgaca gcttacagtg gactaacact cagataggtg

 1081 aatggtcttt acccttgcct ggtggtctaa tggatactgg tgtatttggt caaaatttaa

 1141 gatttcacta tctgtctaga atgggttttt gtgtacacgt acaatgtaat gcatcaaagt

 1201 tccatcaggg tgcattgata attgctatga tccctgaaca tcaaaccccc acacaggttg

 1261 ctaatagttt tgcttatgac cgtgttccca caccaaatca taaatctcag aaccaacaac

 1321 tctgcaacaa tagtataccc gtacactaat tgtacacctg caggttttgg attagcacat

 1381 aattttgtaa cattagttat tagagtatta gtgcccctta gatataatga tggtgcctcc

 1441 acctttgtgc ccataacagt tagtgtagcc ccaatgtgtt cccaatttgc aggcttgaga

 1501 tctgctgtag caaggcaagg tttccctgtt agacaggtac caggaagtca acaatttatg

 1561 acaacacaaa gagacaatgg tatacctata taccctgaat ttgagaagac acatggtttt

 1621 aaactacctg gtagagtgac taacctgcta caagtagccc aagtagggac atttttgaaa

 1681 tttaggaata ataccaatga tgtatgattt ctagtcattt ggttacaacc tatctttctc

 1741 gtttggcaca aatgtatgca aattatagag ggtctgttgt gtttgaattt atgttttgtg

 1801 gcagccaaat ggccactgga aaactgctta ttgcatacac accaccaggt ggctcctcac

 1861 caacaacaag aactgatgca atgttggcaa cacatgttat ttgggacatc ggcctacaat

 1921 caacatgtaa acttgtggtt ccttatatat ctgcatctca gtacagacaa aacaatgtaa

 1981 accaaaccac tttgtcatac aatgggtggg tgactgtgtt ccaacaaaca gcactcgtgg

 2041 tacccccagg tgcaccatca acatgtcaac tggtggtcac tgttagtgca gcagataatt

 2101 ttgtattgag gattcccact gacacaactt actttgcaga ctaccaaggt gacgtcaagg

 2161 atgaggtaca agccagtgtg aacaaagtcc tgcagagtgc gctgaacacc ttacctcagc

 2221 gagaacagtc ttcacaaggg gtcatgataa accaagggaa tgcaccagca ctaacagcgg

 2281 ctgagactgg agaatccgat accaactcag gggggtccac aatggaatta caggcaacaa

 2341 attgtgtttt tagtttgagg gaaacagatt tagaatattt gatgtctagg tattctctta

 2401 tgtatgaaga tagattagat tacactaata gccagggcag taggcatttg agatataact

 2461 tagattttag gacaatagga tattacaaag tttaaggctt ttacatattg gagatttgac

 2521 ttagatgttg tggtaatggt tttggaagat aaaccagctg ctgttaaaaa tcttatgttt

 2581 caagtcctgt atacccctca tggtggagtt gtaccaacca ggcacgactc acgagtttgg

 2641 aatgcaccca attcaactag tgtgtacaca agagtgggaa attgtccagc ttcttttagg

 2701 atacctttta tgtctgtttg caattactat acatcctttt atgatggtga tgggaatttt

 2761 gatcggaatg gtgcctccta tggcattaac ccaggtaatt ttatagggac aatcttttag

 2821 agttaaggta tttcttagac ctgtaaatat agaagcttat atgcccagac ccttaattgc

 2881 ttataaagcc aatggtgatg cagtacaaga tagttcaaca tattatcccg caacacaggc

 2941 aggatattac ccagctaccc agacaggacc atatgagatc tgtcaaacta gaaatgccac

 3001 agagttagtg gaaaccaaat gggccaagta ttcatgctca gtcaaatttg acagaggttc

 3061 atttacagca tggtttgtgg gagaagacct gttattggta ccctaccacg ctgctagcaa

 3121 ttggagtcag acaacacatg tgttcctctg gagagcatgg gagaggaatt ggagggatca

 3181 ccccgaaatg gagatgaaga tccccattgt agacatgtgg acagattcaa ccagggacat

 3241 aacatttctt aaacttgcct atgcaacacc atactggctg gagatgcccc gcaagggctc

 3301 tgcaattgga gaatacgtgg ttgtcgtcaa ttcagcacac tttccctgga agcaatatac

 3361 aggacccaaa ccatttagac acccttactt acacattggt caacataccc agtacagact

 3421 ctggatggca aaaggtgatg ctgataatgg cttctgtgga gccgggttaa tatctaaagg

 3481 gaaactttat ggcatagtta cagctagaac tgagagcaag tcaggggaca tctatgtggc

 3541 ctacaatgaa ttggatgagg atactttcct ccagacacag cagaggtgct ttgattttgg

 3601 aatggataca cacttcaacc tgggaatgca tgactgggtc caaggactcg gccaagtctt

 3661 tggtgagggt gtgtctggag aagttaagaa gcaggtagag gactatctag gccaaattaa

 3721 acccatcata gatgcgggta ccaataaagt taaggatgtt attaaagatg agatggttag

 3781 tgctagcatg tctttattag ttaaagttgt agcttctctt gcttggcgct ttgttaggtg

 3841 tagatatatt tatgactgat ccaataatgt acttgtatag taagattact ggagagccac

 3901 acaaacaagg tccatctgac tggctgaaag attttaaggc caaaggtatg tgcaaaagtg

 3961 cttgaattta agaattgtaa aactacacta ggacaggaag acatttgcca gataaaggtt

 4021 tatatagata aattgattga attgggtaac aaatatggcc ataaatttaa cctgcaaatg

 4081 tctcagttac tgcaatgttc aaacataata aacaaagctt acagtaacat gacaagatct

 4141 agacatgaac ccgtggcaat gcttatacat ggttccccca gatcccaagc attttgatgg

 4201 atataatcaa cagaaagtag ttataatgga tgatttagga caaaacccag atggtgagga

 4261 ttgcaagatg ttttgtcaga tggtgtctac caccacctac ataccaccta tggcttccct

 4321 agatgagaaa ggattaccat ttatttctga ttttgtcttg gcctctacta accaacatgc

 4381 cctgacccct cgcaccatag ctgaaccaga tgctataaat aggagatggt ttatgaatgt

 4441 tgatattcac ctcaagaaag aattcaagga tgacagagga agattggaca tgtcaaagtg

 4501 cttgccttgc aaggattgca aaccaaccaa ctttaggaag tgcaacccat tagtttgtgg

 4561 taaagcaatt attttattag acagaaaaac ccagaaaaat tggactgttg atacagctgt

 4621 tacccactta ttagaagaat ctgaaagaag aaagggtttc ttgaatgtgg ttgatgcgat

 4681 cttccaaggg ccagtgcaga ttccagaatg tgttagggaa gatgaggtga agaagaagaa

 4741 ggtaaattca gagagagata tcccacatga tgtgatggaa ttagttaggt gcacaaagtc

 4801 tcctgtaata atagatgagc ttgagaaggc tggctttatt attcctgtgg aagctgaagt

 4861 tatacgccag accaataatg tgaataatgt aacccaaatt gtttcagcaa ctcttgctag

 4921 tttagctgct ataatttctg taggtactgt agtttattta atggttaggt tattctcctc

 4981 aaaacaaggg gcatactctg gtgcacccag accagaaaca aggaaacctg tgctcaggaa

 5041 agctgtagtt cagggtcctg acatggaatt cgccaagtcc ataatgagat ctaacttgtg

 5101 tcaggttact acaagtgtgg gacctttcac tggacttggt atctatgaca acatcttagt

 5161 gctaccaaga catgcctatg taagtggaaa tgtagtaata gatggtgttg atattcctgt

 5221 tatagatgca gtagaattag aggcagagga aggaaattta gaattagtac agctaaccct

 5281 taagaggaat gaaaagttta gagatattag aaagtttttg agcaatggct ttcatagtga

 5341 gaatgattgt tggctttgca taaattctga gatgttttct aatgtgtaca tacctcttaa

 5401 gaatgtgtct gcctttgggt ttcttaacct gtctatgact cccacctata ggacccttgt

 5461 ttataattac cccaccaaga tgggaaaata taagggcaat gttgatgcta cattaccaga

 5521 agaagccctc atagctattg atcatttggt gtccaaattt aaagcaatag tgccagacaa

 5581 tttgaccgag aagatgtcat tcaatcacta tctcctctta ttaaggcttt agatctttat

 5641 ggatatgatt tacctttcac cacttacctt aaggatgagt taagaccaaa agaaaaagtg

 5701 aagatgggca aaaccagggt cattgagtgt tcatcgctta atgataccat aatgatgaag

 5761 caaactttcg gccacctgtt ccagacatgt cacaagaatc ctggaactta cactggtgtg

 5821 gccgttggct gcaatccaga tgttgactgg tcaaagtttg ctgctgaaat tggtgatgcc

 5881 tatgtttgtg cttttgatta caccaactgg gatgccagcc tgtcacc

//

---

## [Editor Report · Decision Letter 1]

4 Nov 2021

Detection of porcine enteric viruses (Kobuvirus, Mamastrovirus and Sapelovirus) in domestic pigs in Corsica, France.

PONE-D-21-15693R1

Dear Dr. Lisandru Capai

We’re pleased to inform you that your manuscript has been judged scientifically suitable for publication and will be formally accepted for publication once it meets all outstanding technical requirements.

Kind regards,

Humberto Rodney Colina Muñoz, PhD

Academic Editor

PLOS ONE

Additional Editor Comments (optional):

All the questions and suggestions made by the reviewers have been correctly answered and added by the authors.
---

## [Editor Report · Acceptance letter]

5 Jan 2022

PONE-D-21-15693R1 

Detection of porcine enteric viruses (Kobuvirus, Mamastrovirus and Sapelovirus) in domestic pigs in Corsica, France 

Dear Dr. Capai:

I'm pleased to inform you that your manuscript has been deemed suitable for publication in PLOS ONE. Congratulations! Your manuscript is now with our production department. 

Kind regards, 

on behalf of

Dr. Humberto Rodney Colina Muñoz 

Academic Editor

PLOS ONE